Web application firewall based on machine learning models

Durmuşkaya Muhammed Ersin m.durmuskaya@iku.edu.tr 1
Bayraklı Selim 2
1 Department of Computer Engineering, Istanbul Kultur University , Istanbul , Turkey
2 Department of Computer Engineering, Turkish Air Force Academy, National Defence University , Istanbul , Turkey
Chicco Davide
Electronic publication date: 2025 Jul 16
Publication date: 2025
Volume: 11
Electronic Location ID: e2975
Received 2024 Dec 6; Accepted 2025 May 29
Copyright: ©2025 Durmuşkaya and Bayraklı
Copyright year: 2025
Copyright holder: Durmuşkaya and Bayraklı
License: This is an open access article distributed under the terms of the Creative Commons Attribution License, which permits unrestricted use, distribution, reproduction and adaptation in any medium and for any purpose provided that it is properly attributed. For attribution, the original author(s), title, publication source (PeerJ Computer Science) and either DOI or URL of the article must be cited.
License URL: https://creativecommons.org/licenses/by/4.0/

Keywords: Web application firewall, Machine learning, Classification, Web security, WAF, Injection

Funding: The authors received no funding for this work.

==============================
The increasing reliance on web applications for storing sensitive data and financial transactions has elevated the importance of web application security. A machine learning-based web application firewall was designed to protect web applications against injection vulnerabilities. A hybrid dataset, including CISC 2010, HTTPParams 2015, and real-time Hypertext Transfer Protocol (HTTP) requests, was employed. The study evaluated five classification algorithms—K-nearest neighbors, logistic regression, naïve Bayes, support vector machine, and decision tree—for detecting cross site scripting (XSS), Structured Query Language (SQL) Injection, Operating System Command Injection, and Local File Inclusion attacks. Decision tree was identified as the algorithm with the highest precision, accuracy, recall, F1-score, receiver operating characteristic (ROC), and area under the curve (AUC) values. According to the confusion matrix analysis, the real-time tested web application firewalls (WAF) achieved a remarkably high F1 score of 93.13% and accuracy of 93.27%. The findings indicate that machine learning-based WAFs effectively protect web applications against injection threats. Future work includes expanding the WAF to cover other attack types and testing it on different datasets.

Introduction

In the field of cybersecurity, intrusion detection systems (IDS) and intrusion prevention systems (IPS) function as critical safeguards against unauthorized network traffic. These systems, implemented as hardware or software, continuously monitor network traffic. Upon detecting potential security threats, they promptly notify the network administrator, enabling timely intervention (Hock & Kortis, 2015). Web application firewalls (WAFs) emerged as a specialized form of IDS/IPS designed to address the specificities of the Hypertext Transfer Protocol (HTTP) and the associated attack landscape. Typically, WAFs rely on intricate rule sets, often employing regular expressions, to identify and mitigate a wide range of known input and output validation attacks targeting web applications (Luptk, 2011).

The Open Worldwide Application Security Project (OWASP) Top 10 is a widely recognized reference in web application security. It is a collaborative effort that identifies the most significant security weaknesses impacting web applications, offering a consensus-driven compilation aimed at enhancing awareness among professionals in the field. Organizations are strongly encouraged to adopt the OWASP Top 10 and initiate a process of mitigating these risks within their web applications. Implementing OWASP Top 10 recommendations helps establish a security-focused software development culture, resulting in more secure code (Ghanbari et al., 2015).

Cross-site scripting (XSS) is a vulnerability that enables attackers to inject malicious scripts into web pages, leading to unauthorized actions or unauthorized data exfiltration when executed in a user’s browser. SQL injection involves injecting malicious SQL code into queries, allowing attackers to manipulate databases, access sensitive information, or impersonate users. OS command (OSC) injection exploits an application’s ability to execute system commands, potentially enabling attackers full control over the server and further network exploitation. Local file inclusion (LFI) occurs when an attacker gains unauthorized access to files within the server, exposing sensitive data or executing unintended scripts, which may lead to system compromise. These vulnerabilities are categorized under “A03-Injection”, ranked third in the OWASP Top 10, presenting significant risks to web application security. To mitigate such threats, this study proposes a security solution focusing on the detection and prevention of XSS, SQL, LFI, and OSC attacks. One effective approach to safeguarding web applications is the deployment of WAFs, a specialized subset of IDS and IPS. WAFs employ two primary detection methodologies: signature-based and anomaly-based techniques. Signature-based WAFs identify attacks by matching incoming requests against a predefined database of known attack patterns, while anomaly-based WAFs detect deviations from normal behaviour, enabling them to identify both known and novel threats. Due to their broader detection capabilities, anomaly-based WAFs have gained increased attention in cybersecurity research, offering a more adaptable and proactive defence mechanism against evolving web-based attacks (Shipulin, 2018; Stankovic & Simic, 2010).

WAFs utilize diverse deployment strategies to achieve comprehensive web traffic inspection. The reverse proxy mode positions the WAF as an intermediary, facilitating request analysis and forwarding legitimate traffic to the server while simultaneously mitigating malicious attempts. Conversely, the transparent proxy mode allows direct client–server interaction while the WAF operates invisibly in the background. Both these deployments might necessitate dedicated devices. Alternatively, the host-based mode integrates the WAF as a server module, enabling the analysis of requests prior to reaching the server’s core functionality (Yaacob et al., 2012). The proposed WAF model adopts the reverse proxy paradigm, employing the open-source mitmproxy tool for managing HTTP/HTTPS requests and responses (mitmproxy, 2024). WAFs are essential in modern software engineering, improving the security and reliability of web applications. Traditional rule-based WAFs struggle to detect new attack patterns and require frequent manual updates. In contrast, machine learning-driven WAFs use automated pattern recognition, anomaly detection, and adaptive learning to identify threats more effectively. Integrating these systems into the software development lifecycle supports DevSecOps (development, security and operations) practices by enabling continuous security monitoring and proactive threat prevention. This approach also aligns with shift-left security principles, allowing vulnerabilities to be detected and addressed earlier in development. Unlike many studies that rely on offline or simulated testing, the proposed WAF operates in a live environment, providing a practical and scalable solution for secure web application deployment. This not only strengthens security measures but also reduces the need for post-deployment patches and minimizes incident response efforts.

Machine learning (ML) technologies have become instrumental in cyber defence by automating various techniques, including monitoring, control, threat detection and mitigation, and alarm system activation. This study introduces a novel WAF model leveraging ML algorithms. The model is trained on a hybrid dataset constructed by merging publicly available datasets (CSIC 2010, HTTPParams 2015) with a proprietary dataset generated from real-time attacks. The constructed dataset was utilized in K-nearest neighbours (KNN), logistic regression (LR), naive Bayes (NB), support vector machine (SVM), and decision tree (DT) ML algorithms with a 20% test and 80% training split. ML algorithms were evaluated for each vulnerability using precision, accuracy, recall, and F1-score metrics.

In the context of cyberattack detection and live network monitoring, classical ML algorithms and deep learning (DL) models are characterized by fundamentally different methodological properties and operational implications. Traditional ML techniques rely on manual feature extraction and labeled datasets, offering computational efficiency and low-latency inference. These features make them well-suited for real-time security applications, especially in environments with strict time and resource limitations. In contrast, DL models automatically extract features from raw, high-dimensional traffic data and are effective at detecting non-linear patterns in cyber threats. Despite their higher detection accuracy, DL models require substantial computational resources and exhibit increased latency. These factors make them less suitable for resource-constrained, operational WAF deployments. For these reasons, this study employs classical ML methods, prioritizing efficiency, transparency, and adaptability to operational constraints commonly encountered in active web traffic conditions.

This research aims to investigate the following research questions:

Q1. To what extent can traditional ML algorithms effectively detect injection attacks in real-time web traffic?

Q2. Which algorithm among KNN, LR, NB, SVM, and DT demonstrates the most reliable performance for WAF implementation?

Q3. How does the proposed WAF perform in live environments compared to state-of-the-art approaches in the literature?

To address these research questions, this study makes the following contributions:

1. Real-time WAF testing: A method for live testing of the proposed WAF model is presented. While many existing studies develop WAF models, they often lack real-time evaluation in a web environment, which limits their practical applicability in real-world scenarios.

2. Validation of traditional techniques: Experimental results confirm that traditional ML techniques achieve high performance in detecting malicious web requests.

3. Comprehensive up-to-date dataset: A comprehensive and up-to-date dataset of live HTTP traffic has been created to improve the training of ML models. Unlike widely used datasets such as CSIC 2010 and HTTPParam 2015, which do not include modern attack types, the proposed dataset covers SQL injection, XSS, LFI, and OSC injection vulnerabilities. Since it consists of real-time attacks targeting web applications, it provides better adaptability to emerging threats.

4. Evaluation of traditional ML under live condition: While DL-based IDS models are commonly proposed in the literature, they must be tested in active operational settings to validate their effectiveness.

5. Broader scope of attack coverage: Most existing research focuses primarily on detecting XSS and SQL injection attacks. In contrast, this study extends security coverage by incorporating LFI and OSC vulnerabilities, which are critical for injection attacks. This broader scope enhances the proposed security model, making it more comprehensive than conventional approaches in the literature.

These contributions directly address critical gaps in the existing literature by incorporating live performance evaluation, broadening attack coverage, and systematically validating the effectiveness of traditional ML techniques in practical web security applications.

The subsequent sections of this article are organized as follows: ‘Related Work’ provides a critical review of relevant literature, exploring similar studies conducted by other researchers. A detailed description of the methodology and experimental setup is presented in ‘Materials and Methods’. In ‘Results’, the outcomes are described and discussed. Practical implications, limitations of study, further work, and conclusions are described in ‘Conclusions’.

Related Work

WAFs function as essential security safeguards against cyberattacks targeting web applications. They employ diverse detection and blocking techniques for malicious traffic, categorized as signature-based and anomaly-based methods. Signature-based methods: These techniques compare incoming traffic against a database of known attack signatures, effectively mitigating well-established threats. However, their vulnerability lies in their inability to recognize novel, “zero-day” attacks. Anomaly-based methods: Leveraging ML and artificial intelligence, these methods identify normal system behaviour and subsequently detect deviations therefrom, indicating potential anomalies. While capable of recognizing zero-day attacks, they may also generate false positives.

Isiker & Sogukpinar (2021) introduce a WAF situated within anomaly-based categories, leveraging natural language processing (NLP) methodologies alongside a linear SVM learning model. The ML methods used in this study include SVM, DT, RF, KNN and LR algorithms. To evaluate the effectiveness of different NLP approaches, the researchers constructed distinct models based on word n-grams and character n-grams. Their investigation revealed that the character n-gram model outperformed prior research efforts in terms of detection accuracy. They attained the proposed model’s effectiveness in detecting web attacks, achieving an overall accuracy of 99.53%. Guven et al. (2022) employed various ML models, including RF, DT, LR, and NB to achieve multi-class detection of network traffic anomalies. An evaluation was conducted utilizing a pre-processed version of the CICIDS2017 dataset to assess the performance of these ML algorithms. The RF classifier achieved the most favourable outcome, exhibiting a remarkable accuracy of 99.94%. However, Isiker & Sogukpinar (2021) and Guven et al. (2022) rely solely on benchmark datasets for training and testing.

Building upon ML and feature engineering techniques, Shaheed & Kurdy (2022) introduced a novel WAF model for the detection of prevalent web attacks. This model leverages features extracted from HTTP requests, including URL, payload, and header information, to distinguish between normal and anomalous requests. A man-in-the-middle (MITM) proxy served as the parsing tool for HTTP requests. The proposed model focuses on analysing specific features of incoming requests, including request length, character composition (percentage of allowed and special characters), and a measure of potential attack severity (attack weight). To assess the effectiveness of these features, the researchers employed a multifaceted evaluation strategy. This strategy utilized publicly available datasets CSIC 2010, HTTPParams 2015, and Hybrid dataset along with real-world logs captured from a compromised web server. Furthermore, they implemented four distinct classification algorithms (NB, LR, DT, and SVM) alongside two validation techniques (train-test split and cross-validation) to mitigate the overfitting. Román-Gallego et al. (2023) investigated the performance of ML techniques for detecting malicious web requests. Their study explored the performance of various algorithms, including NB, KNN, SVM, and LR. These algorithms were evaluated using a synthetic dataset exceeding 100,000 requests specifically designed to emulate malicious activity. To facilitate the development of their AI-based WAF, they curated a comprehensive dataset encompassing diverse request types. Leveraging publicly available datasets on SQL injection, XSS, OSC injection, and path traversal (referred as LFI) injections from Shah (2019, 2021), Kaggle, and Morzeux-github, the authors constructed a robust resource for training and evaluating their proposed model. Evaluation of the proposed AI-based WAF revealed promising performance across various injection attack types. Notably, the SVM model achieved the highest overall performance, attaining F1-scores of 0.97 for Command Injection and Path Traversal, and a perfect score of 1.0 for both SQL Injection and XSS. Shaheed & Kurdy (2022) and Román-Gallego et al. (2023) attempt to expand attack coverage using hybrid datasets, they do not include active web traffic testing.

Ismail et al. (2023) conducted a comprehensive evaluation of eight supervised ML algorithms for their efficacy in identifying web application assaults. The study compared KNN, NB, LR, SVM, DT, RF, AdaBoost, and artificial neural network (ANN) algorithms. Analysing the CSIC HTTP dataset, a widely recognized standard for evaluating performance in the field, the results revealed an impressive accuracy of 89% and an area under the curve (AUC) of 94% for the KNN and RF classifiers. Particularly noteworthy, the NB classifier emerged as the most computationally efficient option, demonstrating its ability to distinguish malicious from benign HTTP requests with minimal resource utilization. Kassem’s study (2021) aimed to develop an intelligent system that utilizes ML models to detect cyber intrusions. To this end, a host-based intrusion detection system (HIDS) was constructed with text mining methods. To support the system, a validated text dataset encompassing 6,000 records of malicious URLs was created. The nature of this data led to the adoption of the DOC2VEC model as a feature representation technique in the HIDS. The study employed various ML algorithms, including KNN, DT, SVM, and multilayer perceptron (MLP). The results demonstrated that the MLP was the most effective model with an accuracy of 90.67% in detecting directory traversal (similar to LFI), XSS, and SQL injection vulnerabilities. The evaluation of the implemented HIDSs involved different measurements such as confusion matrix, recall, detection rate, precision, accuracy, and area under the receiver operating characteristic (ROC) curve. In terms of detecting Path-Traversal attacks, the SVM algorithm attained the highest accuracy rate, reaching 95.41%. When the DT model was used, the precision and recall rates for the Path-Traversal attack were determined as 89.20% and 82.14%, respectively. The MLP model exhibited a lower precision rate (87.47%) for the Path-Traversal attack. With SVM, the precision and recall rates for the Path-Traversal attack were 75.71% and 95.41%, respectively. Finally, the KNN model yielded a precision rate of 85.11% and a recall rate of 91.84% for the Path-Traversal attack. Models proposed by Kassem (2021) and Ismail et al. (2023) demonstrate strong results in static and host-based environments but lack deployment and validation in dynamic web traffic scenarios.

In their work, Habibi & Surantha (2020) provide a concise overview of web application attack detection methods, specifically focusing on identifying XSS vulnerabilities. They explore the application of NB, KNN, and SVM algorithms with the n-gram method, a model designed to uncover similarities between text sequences. In contrast, Harish Kumar & Godwin Ponsam (2023) employ a broader range of classification algorithms for web application attack detection. While they also utilize NB, KNN, and SVM ML algorithms, their research further incorporates DT, LR, and RF algorithms. They also leverage DL models such as convolutional neural networks (CNN) and long short-term memory (LSTM), in conjunction with boosting algorithms like Gradient Boosting and AdaBoost. According to Kascheev & Olenchikova (2020), an experiment was conducted to develop a technique for XSS attack detection. The research estimated the performance of various ML models, including SVM, LR, DT and NB classifiers. The author emphasizes that evaluating an algorithm’s results using a single metric is not an impartial approach. Therefore, the study employed a set of metrics (F-measure, precision, accuracy, and recall) to assess the algorithms’ performance comprehensively. Among the measured models, the DT model exhibited the recall of 93.70%, an accuracy of 98.81%, and a precision of 99.19%. Habibi & Surantha (2020), Harish Kumar & Godwin Ponsam (2023), and Kascheev & Olenchikova (2020) investigate various ML algorithms, primarily focusing on XSS detection, while placing limited emphasis on deployment feasibility.

Shar, Briand & Tan (2015) investigate the application of supervised ML models for vulnerability prediction, focusing specifically on identifying remote code execution and file inclusion vulnerabilities. They employ RF and LR classifiers and estimate their performance using two validation techniques: cross-validation and cross-project validation. Their evaluation metrics include recall, precision, and probability of false alarm. Notably, the recall values achieved by the LR and RF models for file inclusion detection are 81% and 76%, respectively. Tao & Meng’s (2022) experimental evaluations (from real campus network traffic data) demonstrate that the proposed method achieves a classification accuracy of up to 95.3%. This indicates its effectiveness in detecting network traffic containing file inclusion attacks, offering a promising new approach for web security. Prabakaran et al. (2022) propose a Software-Defined Network Function Virtualization (SDNFV) framework that integrates ML-based attack prediction with a stateful firewall VNF to enhance network security. Their study focuses on detecting denial-of-service (DoS) attacks in Software-Defined Networking (SDN) environments, utilizing Bayesian network, NB, C4.5, and Decision Table algorithms, with Bayesian Network achieving the highest accuracy of 92.87%. While their model is trained on historical intrusion data from AWS honeypots, its real-world applicability remains uncertain, as evaluations were conducted in a simulated Mininet SDN environment. Their findings emphasize the potential of ML-driven security mechanisms for network protection. While Tao & Meng (2022) utilize real network traffic, their work remains limited to file inclusion detection, without addressing broader attack categories or deployment in a live WAF environment. Similarly, Prabakaran et al. (2022) contribute to intrusion detection frameworks, but their efforts are confined to host-based contexts and lack WAF-specific validation.

Automated security testing plays a crucial role in evaluating the effectiveness of WAFs. Liu, Li & Chen (2019) introduced the first approach that formulates WAF test case generation as a search problem, leveraging semantic learning to navigate the search space efficiently. Their method employs differential evolution (DE) to generate security test cases, demonstrating significant improvements in detecting SQL injection vulnerabilities. Extending this work, Li, Yang & Visser (2023) proposed DANUOYI, an evolutionary multi-task injection testing framework that simultaneously generates test cases for multiple injection attack types, including SQL, XML, and PHP injections. By utilizing multi-task learning and cross-lingual translation models, DANUOYI enhances test input diversity and discovers vulnerabilities that single-task approaches fail to detect. These contributions provide a solid foundation for WAF evaluation methodologies and complement ML-based WAFs by enabling more comprehensive robustness assessments.

Recent advancements in security and robustness for software engineering problems have increasingly incorporated genetic algorithms and ML to enhance vulnerability detection and mitigation. Zhou et al. (2024) propose an evolutionary multi-objective optimization framework for generating adversarial examples in software engineering, leveraging CodeT5 to create context-aware perturbations that can expose weaknesses in DL models. This approach highlights the role of evolutionary techniques in refining security mechanisms against adversarial threats. Additionally, Zhou, Li & Min (2022) introduce a genetic algorithm-based adversarial attack method, the attention-based genetic algorithm, which employs an attention mechanism to improve the efficiency of adversarial perturbations in natural language processing (NLP) tasks. This technique underscores the necessity of evolutionary strategies in fortifying ML-based security models. Furthermore, Feng et al. (2024) explore the integration of genetic algorithms with prompt learning to optimize pre-trained language models for code intelligence tasks, demonstrating improved defect prediction and code summarization accuracy. These studies collectively reinforce the significance of evolutionary computing in addressing security vulnerabilities, providing a robust foundation for enhancing the detection capabilities of ML-driven WAFs. Studies by Liu, Li & Chen (2019), Li, Yang & Visser (2023), and Zhou et al. (2024) highlight advances in automated WAF testing and adversarial robustness, yet they primarily focus on input generation or model vulnerability rather than real-time deployment.

In summary, existing literature provides valuable insights into the application of ML for web attack detection. However, many studies rely on synthetic or static datasets and lack real-time evaluation. While several approaches demonstrate high classification accuracy, only a few attempt runtime testing in live environments or address broader attack vectors. This reveals a gap in operational validation, which is crucial for the practical adoption of WAFs in dynamic web settings.

Materials & Methods

The ML-based WAF, implemented in Python, mediates between the web server and clients. The source code for the ML-based WAF tool is publicly available on GitHub (Durmuskaya, 2024). The application receives HTTP/HTTPS requests, analyzes their structure (parses them) and extracts features to make compatible for ML algorithms. Subsequently, the WAF classifies the requests based on whether they contain vulnerabilities and blocks the malicious requests (Fig. 1).

Figure 1 Integration of the proposed machine learning-based waf into a standard web server architecture.

In the proposed WAF model, feature extraction was performed using the dataset, followed by an evaluation of different ML algorithms. The comparison of KNN, NB, LR, SVM, and DT aimed to assess the performance of commonly used, computationally efficient, and fast classification models in security applications. Among these, the DT algorithm was identified as the most effective due to its high performance and interpretability in detecting malicious web requests, making it the optimal choice for operational WAF deployments.

KNN is a non-parametric algorithm that classifies instances based on the majority class among the k nearest neighbors. It performs effectively when class boundaries are well-defined, and the dataset is relatively small. SVM construct an optimal hyperplane to separate data classes with the maximum margin. By leveraging kernel functions, SVMs are capable of handling both linearly and non-linearly separable data, and they offer robustness against overfitting, even with relatively small training sets. DT split data recursively based on feature values to form interpretable rule-based models. NB is a probabilistic classifier that assumes feature independence, offering computational efficiency and strong performance in high-dimensional applications like spam detection. LR is a statistical classification technique that models the probability of categorical outcomes. It is suitable for both binary and multiclass classification problems and assumes a probabilistic relationship between independent variables and the target class.

Proposed WAF model in a real-time environment

The DT model, which demonstrated superior performance, was integrated into the WAF system to enhance its functionality. The model was trained on the dataset, ensuring adaptability to various attack types. In active deployment, the WAF listens and parses HTTP requests using mitmproxy, while the trained DT model classifies incoming traffic as either anomalous or normal. Any request identified as anomalous is logged and blocked, preventing potential web security threats (Algorithm 1).

Algorithm 1 Proposed WAF algorithm pseudocode.

PRELIMINARY server_ip_address(ip1), mitmproxy_ip_address(ip2), server_port(p1), mitmproxy_port(p2), hybrid data set(d),	
(1) START	
(2) SET input (d, p1, ip1, p2, ip2)	
(3) PREPROCESSİNG for d (missing values, URL decoding,scaling)	
(4) COUNT alphanumeric char. and “badwords” IN d	
(5)       CONVERT d to CSV file	
(6) SET WAF inputs (Model: Decision Tree, test and train data FROM csv file)	
(7) WHILE mitmproxy listener is “ON”	
(8)       WAIT FOR HTTP requests	
(9)       IF HTTP request come	
(10)             PARSE HTTP request to PATH, CONTENT, HEADER	
(11)             COUNT alphanum. char., badwords from Path, Content, Header	
                              SAVE to csv file	
(12)             CHECK the counting element in CSV file WITH WAF (decision tree)	
(13)             IF the request is “VALİD”	
(14)                   ALLOW http traffic	
(15)             ELSE	
(16)                   DROP request	
(17)                   PRINT “403, forbidden”	
(18)                   SAVE traffic to csv file (logging)	
(19)             END IF	
(20)       END IF	
(21) END WHILE	
(22) END	

A graphical user interface (GUI) was developed using the Tkinter module within Python to provide user control over the WAF. Toggle buttons enabled the activation and deactivation of the WAF, while a dedicated text box presented a real-time visualization of all HTTP requests received by the application. Using a color-coding scheme, potentially malicious requests are highlighted with a yellow background in the text box, facilitating immediate identification and response (Fig. 2).

Figure 2 Graphical user interface of the proposed machine learning-based web application firewall.

Mitmproxy was employed to manage web traffic through the WAF. As a Python-based proxy tool, Mitmproxy facilitates the execution of Python scripts. The implemented script enabled the collection of specific fields (host, path, and content) from web traffic for further analysis. In this context, “host” refers to the domain or IP address of the web server receiving the request, “path” represents the specific resource or endpoint being accessed within the web application, and “content” includes the body of the request, which may contain user inputs or payloads. Furthermore, ML capabilities were integrated for logging, live detection, and prevention of malicious requests. This resulted in the establishment of a ML-driven WAF. The developed WAF model provides security against vulnerabilities such as LFI, XSS, SQL injection, and OSC injection. To evaluate the WAF’s effectiveness, datasets were prepared for each of these vulnerabilities and provided as training and test sets for the KNN, SVM, NB, LR, and DT algorithms. Table 1 shows the F1-scores for each web application type. The DT classifier has the highest F1-score for these vulnerabilities. The choice of the DT algorithm for the WAF is based on its superior predictive performance for the targeted vulnerabilities.

Table 1 Performance comparison of machine learning algorithms across different injection vulnerabilities (SQL, XSS, LFI, OSC) based on F1-score.

ML algorithms	F1-score for different vulnerabilities	
	XSS	SQL	OSC	LFI	
Logistic regression	0.9944	0.7255	0.9788	0.9245	
Naive Bayes	0.9954	0.8500	0.6937	0.4457	
KNN	0.9962	0.9858	0.9818	0.7190	
SVM	0.9951	0.9952	0.9800	0.9376	
Decision Tree	0.9965	0.9957	0.9818	0.9381	

An open-source web application security testing platform, DVWA, was deployed on a virtual machine utilizing XAMPP. To achieve this, the XAMPP framework, known for facilitating web server environment creation and operation, was employed to launch the DVWA application on the localhost, leveraging the Apache web server and MySQL database management system. Furthermore, on this virtual machine, a custom Python script designed for mitmproxy was executed to assess the WAF’s accuracy. This implementation enabled the monitoring of web traffic and the subsequent blocking of malicious requests (Fig. 3).

Figure 3 Architecture of the experimental environment used for testing the proposed machine learning-based waf.

On the host machine (attacker side), attacks were directed towards the DVWA application within the virtual machine using web-exploitable payloads. To automate these attacks, Burp Suite was used, evaluating the WAF’s efficacy. Through this assessment, the WAF’s accuracy and F1-score were determined to be 93.27% and 93.13%, respectively.

Data set

Two distinct datasets were employed to model malicious HTTP requests. The first dataset, a self-generated collection (Durmuskaya, 2024), has been made publicly available on Figshare to support replication and further validation of our approach. This dataset was constructed using a comprehensive set of payloads obtained from various GitHub repositories, including SecLists, Offensive-Payloads, PayloadsAllTheThings, payloadbox, danielmiessler, and omurugur. These payloads were utilized in automated attack simulations against vulnerable web applications (demo.testfire.net, testphp.vulnweb.com, bWAPP, and juice-shop.herokuapp.com), with Burp Suite facilitating the attack execution. The resulting attack logs were parsed and transformed into features suitable for integration within ML algorithms. Further enriching the modelling process, a second dataset, composed of the CSIC 2010 dataset (Giménez, Villegas & Marañón, 2010) and the HTTPParams 2015 dataset (Morzeux, 2020)—as referenced in Shaheed & Kurdy (2021)—was also incorporated. The integration of the two employed datasets resulted in the construction of a labelled the proposed dataset. The performance of various ML algorithms was evaluated using the dataset. The classification algorithm that demonstrated the most promising results was selected (Fig. 4).

Figure 4 Workflow for selecting the optimal classification algorithm for the proposed machine learning-based waf.

To record normal HTTP requests, Burp Suite’s web crawling feature was utilized. This feature automatically discovers and crawls all linked URLs starting from a specified root URL, generating a comprehensive list of benign HTTP requests. The log file from this crawl contains information such as the date and time of each request, the URL, the HTTP method, headers, and body. This log of normal traffic was then parsed into a format suitable for ML. Table 2 presents the distribution of HTTP request counts within the dataset.

Table 2 Distribution of benign and malicious HTTP requests extracted from burp suite log files for model training.

Vulnerabilities	Abnormal requests	Normal request	
XSS	29,000	20,000	
SQL	20,000	20,000	
OSC	21,000	20,000	
LFI	41,000	20,000	

During data collection, benign requests were gathered only from publicly accessible pages of the target website, ensuring that only information available to a regular user was collected. No unauthorized access or exploitation of vulnerabilities was attempted, in adherence to ethical guidelines. For malicious requests, the web applications tested were specifically designed for security research and training. These platforms explicitly permit such security assessments, ensuring no ethical standards were violated. The study strictly followed ethical principles, avoiding any unauthorized intrusion or harm to real-world systems.

Data processing

Normal and anomalous HTTP requests were recorded in a log file using the Burp Suite program. The file’s contents, depicted in Fig. 5, reveal that HTTP requests are encoded using the base64 format, a binary-to-text encoding scheme that converts binary data into an ASCII string representation. Once decoded, the “path” and “content” segments of the HTTP requests are visible in plaintext.

Figure 5 Sample from Burp Suite log file showing captured HTTP requests used for dataset generation.

Table 3 presents the selected features for each attack type, incorporating both a predefined list of “badwords” and the special characters analysed through statistical methods in Table 4. These features were identified based on their frequency and relevance in distinguishing between benign and malicious requests. The “badwords” list consists of commonly used terms in attack payloads, while the special characters include symbols such as single and double quotes, hyphens, dashes, slashes, and backslashes, which are often associated with web-based vulnerabilities.

Table 3 List of feature tokens: badwords and special characters extracted for malicious request detection.

Attack type	Badwords list	Special characters	
XSS	script, alert, onload, string, fromcharcode, meta, input, type, button, action, iframe, javascript, onmouseover, document, onerror, confirm, formaction, newline, tab, svg, onafterprint, onbeforeprint, onbeforeunload, onhashchange, onmessage, ononline, onoffline, onpagehide, onpageshow, onpopstate, onresize, onstorage, onunload, onblur, onchange, oncontextmenu, oninput, oninvalid, onreset, onsearch, onselect, onsubmit, onkeydown, onkeypress, onkeyup, onclick, ondblclick, onmousedown, onmousemove, onmouseout, onmouseup, onmousewheel, onwheel, ondrag, ondragend, ondragenter, ondragleave, ondragover, ondragstart, ondrop, onscroll, oncopy, oncut, onpaste, onabort, oncanplay, oncanplaythrough, oncuechange, ondurationchange, onemptied, onended, onloadeddata, onloadedmetadata, onloadstart, onpause, onplay, onplaying, onprogress, onratechange, onseeked, onseeking, onstalled, onsuspend, ontimeupdate, onvolumechange, onwaiting, onshow, ontoggle, prompt, src, body, object, title, frameset, style, applet, xml, div, table, base, xss, classid, import, namespace	Character <
Character >
Character –
Character #
Character *
Character ;
Character “	
SQL	or, and, like, having, where, injectx, order, order by, rlike, select, case, when, drop, union, group by, limit, system_user, table_schema, table_name, from, information_schema, tables, substring, sysserverse, sysusers, xp_cmdshell, backup, database, create, table, insert, null, exec, sp_addlogin, sp_addsrvrolemember, sysadmin, mysql.user, connect, char, waitfor, delay, pg_sleep, hex, delete, sleep, nvarchar, benchmark, md5, print, objectclass, sqlvuln, members, load_file, sqlattempt2, nslookup, begin, bfilename, replace, count, tabname, syscolumns, selectchar, convert	Character *
Characters –
Characters ——
Characters &&
Characters __
Characters /*
Character @
Character ‘	
OSC	type, necho, usr, bin, whoami, ipconfig, system, cat, phpinfo, exec, phpversion, pwd, eval, echo, sleep, curl, wget, which, netstat, dir, uname, nid, perl, systeminfo, reg, print, netsh, hexdec, dechex, sysinfo, net, cmd, server, route, ping, ifconfig	Characters –
Character —
Characters &&
Character $
Character <
Character >
Character !	
LFI	etc, passwd, zxrj, l3bhc3n3za==, li4v, shadow, aliases, anacrontab, apache2, at.allow, at.deny, bashrc, bootptab, hosts, httpd, opt, proc, root, usr, lib, local, sbin, var, adm, mysql, atfp_history, bash, ssh, boot.ini, c:\, localstart.asp, apache, volumes, c:/, desktop.ini, programfiles, xampp, bin, winnt, conf, cmdline, nginx, database, hostname	Characters ../
Characters ..\\
Characters .\\.
Characters ..\\..
Characters ....\\
Characters ..../	

Table 4 Statistical analysis of feature set for malicious and benign http request classification.

Attack type	Character	PMI	Character	Chi2 score	Character	Mutual information	
XSS	Character (	0.37	Character ”	193,380.46	Character ”	0.61	
Character )	0.37	Character &	51,827.56	Character {	0.60	
Character {	0.16	Character .	25,423.47	Character }	0.60	
Character }	0.16	Character -	22,754.85	Character <	0.58	
Character @	0.16	Character _	21,895.52	Character >	0.57	
Character —	0.00	Character >	21,285.98	Character )	0.53	
Character !	−0.09	Character <	21,069.70	Character (	0.53	
Character *	−0.55	Character ?	11,042.93	Character /	0.43	
Character ;	−0.81	Character }	8,538.31	Character .	0.34	
Character <	−0.84	Character {	8,536.20	Character =	0.27	
Character >	−0.87	Character )	8,497.26	Character ?	0.25	
Character ’	−1.10	Character (	8,494.71	Character &	0.16	
Character =	−1.28	Character =	7,949.08	Character _	0.14	
Character \	−1.64	Character +	4,580.36	Character -	0.07	
Character #	−1.65	Character ;	1,164.45	Character ;	0.03	
Character [	−1.66	Character /	895.66	Character @	0.02	
Character /	−2.04	Character ’	545.59	Character ]	0.01	
Character ]	−2.27	Character [	461.15	Character [	0.01	
Character -	−2.84	Character —	298.07	Character ’	0.01	
Character .	−4.00	Character *	87.50	Character +	0.00	
Character +	−4.09	Character #	72.15	Character #	0.00	
Character ”	−4.35	Character @	70.33	Character !	0.00	
Character ?	−4.66	Character \	60.03	Character \	0.00	
Character &	−6.60	Character ]	44.19	Character *	0.00	
Character _	−7.45	Character !	0.96	Character —	0.00	
SQL	Character #	0.59	Character ”	15,736.83	Character =	0.48	
Character *	0.25	Character /	13,871.10	Character &	0.35	
Character [	0.00	Character -	10,036.19	Character /	0.30	
Character .	−0.42	Character ’	8,682.91	Character +	0.29	
Character ;	−0.42	Character +	7,984.51	Character _	0.14	
Character ?	−0.43	Character )	7,251.04	Character ’	0.13	
Character @	−0.52	Character (	6,337.72	Character )	0.12	
Character _	−0.53	Character —	1,664.68	Character .	0.12	
Character )	−0.59	Character .	1,557.64	Character -	0.09	
Character !	−0.64	Character [	1,181.58	Character (	0.09	
Character ’	−0.67	Character ]	1,174.80	Character ”	0.05	
Character +	−0.68	Character {	1,152.43	Character #	0.05	
Character (	−0.74	Character }	1,151.03	Character ?	0.04	
Character —	−1.01	Character =	1,133.97	Character *	0.03	
Character /	−1.09	Character *	985.22	Character —	0.02	
Character \	−1.33	Character #	904.49	Character @	0.02	
Character =	−2.08	Character ?	828.32	Character ;	0.02	
Character -	−2.16	Character _	443.77	Character [	0.01	
Character >	−2.20	Character &	295.73	Character ]	0.01	
Character &	−2.24	Character >	154.28	Character {	0.00	
Character <	−2.64	Character @	130.44	Character }	0.00	
Character ”	−2.85	Character <	127.01	Character \	0.00	
Character }	−7.92	Character \	70.61	Character !	0.00	
Character {	−7.92	Character ;	36.10	Character >	0.00	
Character ]	−7.95	Character !	24.96	Character <	0.00	
OSC	Character *	0.63	Character ”	19,376.14	Character &	0.52	
Character #	0.52	Character .	16,504.00	Character =	0.51	
Character —	0.44	Character -	15,541.65	Character .	0.39	
Character ’	0.33	Character _	12,131.46	Character —	0.30	
Character [	0.13	Character )	9,249.47	Character ?	0.28	
Character ]	0.13	Character (	9,229.76	Character /	0.23	
Character +	0.06	Character —	6,872.78	Character )	0.18	
Character \	−0.09	Character ?	6,470.08	Character (	0.18	
Character {	−0.51	Character &	4,724.82	Character _	0.13	
Character (	−0.51	Character =	3,606.12	Character +	0.12	
Character )	−0.51	Character /	3,312.42	Character -	0.08	
Character }	−0.56	Character ’	1,881.39	Character ”	0.06	
Character ;	−0.78	Character ;	1,338.53	Character ’	0.05	
Character &	−1.07	Character \	823.69	Character [	0.04	
Character /	−1.47	Character @	618.47	Character ]	0.04	
Character =	−1.49	Character #	510.21	Character ;	0.04	
Character <	−1.70	Character {	348.36	Character {	0.04	
Character >	−1.91	Character }	317.63	Character }	0.04	
Character ”	−2.81	Character <	95.24	Character \	0.03	
Character -	−2.84	Character !	93.51	Character #	0.03	
Character !	−2.93	Character >	69.48	Character !	0.00	
Character .	−4.73	Character [	25.87	Character >	0.00	
Character _	−5.58	Character ]	24.99	Character <	0.00	
Character ?	−5.90	Character +	12.77	Character @	0.00	
Character @	−6.51	Character *	9.89	Character *	0.00	
LFI	Character !	0.00	Character &	66,352.24	Character =	0.41	
Character )	0.00	Character ”	51,577.92	Character /	0.32	
Character ”	0.00	Character -	44,103.54	Character &	0.17	
Character ’	0.00	Character .	40,331.87	Character ?	0.17	
Character ]	0.00	Character =	24,957.93	Character -	0.15	
Character @	0.00	Character _	24,320.78	Character _	0.12	
Character [	0.00	Character \	18,942.39	Character .	0.11	
Character (	0.00	Character +	6,808.96	Character \	0.04	
Character ?	−0.26	Character /	6,584.66	Character }	0.01	
Character }	−0.31	Character ?	3,343.55	Character {	0.01	
Character {	−0.31	Character [	2,270.79	Character ]	0.01	
Character =	−0.78	Character ]	2,270.79	Character [	0.01	
Character *	−1.54	Character @	1,281.86	Character ;	0.01	
Character ;	−1.61	Character ;	1,095.47	Character +	0.01	
Character /	−2.40	Character (	556.28	Character ”	0.00	
Character \	−3.33	Character )	556.28	Character ’	0.00	
Character .	−3.41	Character >	415.35	Character (	0.00	
Character #	−3.81	Character #	400.91	Character )	0.00	
Character <	−4.41	Character —	336.64	Character !	0.00	
Character _	−4.51	Character <	328.79	Character >	0.00	
Character >	−4.94	Character ’	274.11	Character @	0.00	
Character —	−6.54	Character !	233.80	Character <	0.00	
Character +	−7.09	Character {	198.02	Character —	0.00	
Character -	−7.34	Character }	197.09	Character #	0.00	
Character &	−12.55	Character *	4.64	Character *	0.00	

To evaluate the significance of the selected features, three statistical techniques—chi-square, pointwise mutual information (PMI), and mutual information (MI)—were applied to a dataset consisting of approximately 80,000 HTTP requests, encompassing both benign and malicious samples across four different attack types: XSS, SQL Injection, OSC Injection, and LFI. The analysis involved computing the frequency of special characters listed in Table 4, followed by statistical evaluations using chi-square, PMI, and MI. The results provided a comprehensive assessment of each feature’s discriminative power in differentiating between normal and malicious traffic.

The rationale for employing multiple statistical measures lies in their complementary strengths in feature selection. The chi-square test determines the features most strongly associated with attack classifications, PMI highlights the most informative attributes by analysing their co-occurrence probabilities, and mutual information quantifies the overall dependency between features and class labels. Integrating these three methods ensures a more robust and reliable feature selection process, enhancing the performance of ML models in web application security. To implement the feature selection for the ML-based web security model, the frequency of each selected feature within the “path” and “content” fields of every HTTP request was recorded. These frequencies were structured into distinct columns and saved in a CSV file. Each vulnerability type exhibited characteristic patterns; for instance, SQL injection attacks frequently include single and double quotes, whereas XSS attacks commonly contain special characters such as the “<” and “>” symbols and semicolons. A “good” label was assigned to normal requests and a “bad” label to abnormal requests. After all requests were combined into a single CSV file, label encoding was applied to the last column (converting “good” and “bad” labels to numeric values) for the ML process.

The dataset was split into 20% test and 80% training. Then, the feature scaling methods of normalization and standardization were applied to the samples. Normalization involves rescaling feature values to a predefined range, typically between 0 and 1, to ensure uniform contribution of all features and mitigate the influence of large numerical values. Standardization, on the other hand, transforms the data distribution by adjusting feature values to have a mean of 0 and a standard deviation of 1, thereby improving the compatibility of the dataset with ML algorithms that assume normally distributed input features. StandardScaler and MinMaxScaler (is known as normalization) are calculated as in Eqs. (1) and (2) respectively. (1) xnorm=x−xminx−xmax

(2) z=x−μxσx

where xmin and xmax is feature range; standard score of samples x is z, mu is the mean of the training samples and σ is the standard deviation of the training samples.

Performance evaluation criteria

Following data preprocessing and feature extraction, ML models were trained. Within our proposed WAF model, various classification algorithms were employed and compared, including KNN, LR, NB, SVM and DT. To evaluate the performance of these classifiers, several metrics were utilized, focusing on measures that accurately identify anomalous requests. True negatives (TN), true positives (TP), false negatives (FN), and false positives (FP) were employed to compute assessment criteria such as accuracy, precision, recall, and F1-score. These metrics were calculated using formulas (3), (4), (5), and (6), respectively. (3) Accuracy=TP+TNTP+TN+FP+FN

(4) Precision=TPTP+FP

(5) Recall=TPTP+FN

(6) F1−score=2TPTP+FPTPTP+FNTPTP+FP+TPTP+FN

To assess the generalizability of the model across the entire dataset, a 10-fold cross-validation technique was employed. In 10-fold cross-validation, the data is partitioned into ten equal subsets. Each subset is used once as a test set while the remaining nine serve as training data, and this process is repeated ten times. The primary aim of employing cross-validation in this study was to enhance the reliability and generalizability of the model’s performance metrics. By systematically rotating the test/training splits, this method mitigates the risk of overfitting to any particular subset of data and ensures that the reported performance reflects the model’s ability to generalize to unseen data. Based on this comprehensive evaluation, the DT model demonstrated superior performance. Consequently, the WAF model leverages a ML approach based on DT for effective anomaly detection.

Results

The experiments were conducted on a host machine featuring an Intel® Core™ i7-12700H CPU @ 2.30 GHz and 16 GB RAM. A virtual machine with 6 GB RAM was allocated to run the web server, DVWA application, and the WAF. The application of ML techniques involved the use of the scikit-learn Python library.

The ML models in this study were trained using specific hyperparameters and configurations to evaluate their performance in detecting injection vulnerabilities. LR model was implemented with random_state = 0 to ensure reproducibility, using the default L2 regularization and lbfgs solver, which are effective for binary classification tasks. NB classifier employed in this study was the Gaussian variant (GNB), which assumes normally distributed features and utilizes the default var_smoothing = 1e−9 parameter to enhance numerical stability by adding a small constant to the variance—especially beneficial in high-dimensional input spaces derived from tokenized web traffic. KNN algorithm was configured with n_neighbors = 1 and the Minkowski distance metric, which generalizes both Euclidean and Manhattan distances; this setup effectively captures local decision boundaries and is well-suited for detecting point anomalies. For the SVM, a radial basis function (RBF) kernel was used to learn complex, non-linear decision surfaces in high-dimensional spaces. The C and gamma parameters were retained at their default values (C = 1.0, gamma = ‘scale’) to balance model generalization and fitting capacity. The DT Classifier was trained using the entropy criterion, which selects splits based on information gain, optimizing the model’s ability to distinguish between different classifications. To comprehensively assess the methodology, multiple evaluation metrics were utilized. F1-score was adopted as the primary performance metric due to its effectiveness in balancing precision and recall. This decision was crucial for minimizing both false positives and false negatives. Such balance is particularly important when detecting low-frequency attack types within high-volume traffic. Accuracy, while traditionally used, was deprioritized. In imbalanced scenarios, it can yield misleading results by favoring the majority class. Therefore, it was not considered a reliable measure in this context. The selection of additional metrics was guided by the operational characteristics of each algorithm. DTs are known for learning hierarchical decision boundaries. They demonstrated strong performance across both frequent and rare intrusion patterns. As a result, F1-score and AUC were identified as suitable evaluation metrics for this model. Support vector machines leverage margin-based classification. They benefited from AUC analysis, which effectively captures their discriminative ability. Conversely, Gaussian NB, while computationally efficient, showed limitations in precision. These limitations arise from its assumption of feature independence. This highlights the importance of recall in evaluating such models. These methodological considerations informed our overall metric selection. They ensured a comprehensive and reliable assessment of each model’s performance in real-time WAF scenarios.

Figure 6 presents the F1-scores of the ML models for detecting different vulnerabilities. All models demonstrated impressive performance in identifying XSS attacks. However, for SQL injection detection, the LR and NB models performed worse than the others. The KNN algorithm achieved high scores across most vulnerabilities. For LFI attacks, the NB model had the weakest F1-score among all algorithms. Both SVM and DT consistently showed strong results, with DT slightly outperforming SVM overall. A minor decrease in F1-score was observed for LFI attacks with these models as well. The NB model also exhibited the lowest performance in detecting OSC attacks, with an F1-score significantly lower than those of the other models (which averaged around 98%). In summary, while all models effectively detected XSS attacks, their performance varied more noticeably for SQL, OSC, and LFI vulnerabilities.

Figure 6 F1-score comparison of machine learning algorithms for detecting XSS, SQL injection, OS command injection, and LFI attacks.

The evaluation metrics were computed using probability estimates or decision function values provided by the classification models. LR, Gaussian NB, KNN, and DT classifiers generated probability scores using the predict_proba function, while the SVM used the decision_function, which returns a margin-based score rather than direct probabilities. Before calculating the ROC curve, the target labels were transformed into a binary format using LabelEncoder to ensure compatibility with the roc_curve function. The ROC curve was constructed by computing the false positive rate (FPR) and true positive rate (TPR) at different threshold values, allowing the model’s performance to be visualized across various decision boundaries. A reference diagonal line (y = x) was included in the plot to represent a random classifier. The AUC values were obtained by integrating the ROC curve using the auc function, which calculates the area under the TPR-FPR curve. A higher AUC indicates superior classification performance, demonstrating a model’s ability to distinguish between positive and negative samples effectively.

Figure 7 depicts the ROC curves for XSS, SQL, LFI, and OSC vulnerabilities, respectively. Table 5 presents the AUC values calculated based on these ROC curves for each vulnerability. While the AUC values for these vulnerabilities are similar, the DT algorithm yielded slightly better results. The AUC values for the DT were calculated as 0.9976, 0.9988, 0.9964, and 0.9528 for these vulnerabilities, respectively.

Figure 7 ROC curves of machine learning algorithms for detecting XSS(A), SQL injection (B), OS command injection (C), and LFI (D) attacks.

Table 5 Area under the ROC curve (AUC) scores of machine learning algorithms for injection attack detection.

ML-ALGORITMS	AUC	
	XSS	SQL	OSC	LFI	
Logistic Regression	0.9946	0.7155	0.9955	0.9521	
Naive Bayes	0.9942	0.6872	0.9903	0.9527	
KNN	0.9966	0.9752	0.9864	0.9432	
SVM	0.9974	0.9979	0.9863	0.9517	
Decision Tree	0.9976	0.9988	0.9963	0.9527	

DT algorithm was selected for real-time WAF testing, as it demonstrated the highest performance across the targeted attacks based on AUC and F1-score values. To ensure a more reliable and balanced evaluation of the ML models, a 10-fold cross-validation approach was applied.

Figure 8 presents the F1-score vs. K-folds analysis, illustrating the model’s performance across different proposed attack categories under various cross-validation folds. The results indicate that the optimal K-fold value varies depending on the attack type, emphasizing the need to balance bias and variance for robust model evaluation. The highest F1-score was achieved at K = 2 for XSS detection (0.9969), K = 6 for SQL Injection (0.9940), K = 3 for OSC injection (0.9859), and K = 10 for LFI detection (0.9254). These findings suggest that while a lower K value provided optimal performance for certain attack types, higher values contributed to better generalization, particularly in LFI detection.

Figure 8 F1-score performance across 10-fold cross-validation for XSS, SQL injection, OS command injection, and LFI detection.

The K-fold cross-validation results confirm that the model does not exhibit overfitting, as its performance remains stable across different fold values. This consistency demonstrates that the DT model effectively captures critical attack patterns without excessively tailoring itself to the training data.

To further enhance performance, hyperparameters were optimized specifically for the DT algorithm across the four attack categories. The max_depth, min_samples_split, and min_samples_leaf parameters were tuned to control model complexity and improve generalization. max_depth restricted the tree’s growth, preventing excessive branching and reducing overfitting risk. min_samples_split ensured that nodes were divided only when a sufficient number of samples were available, avoiding unnecessary complexity. min_samples_leaf prevented the formation of overly detailed patterns by requiring a minimum number of observations in each leaf node, thereby maintaining model stability.

Randomized search CV was employed for hyperparameter optimization to improve computational efficiency. The search was conducted within predefined ranges: max_depth from 1 to 8; min_samples_split from 2 to 100 in increments of 10; and min_samples_leaf from 2 to 100 in increments of 5. Additionally, the cross-validation (cv) parameter of the search was set to 100, providing an extremely thorough evaluation by using 100-fold splits (this unusually high number of folds further reduces variance in performance estimates, albeit with a high computational cost). The number of iterations (n_iter) was limited to 10, meaning the search explored 10 random combinations of hyperparameters, which balances search breadth with computational cost. Table 6 summarizes the optimal hyperparameter values obtained for each vulnerability category from this search.

Table 6 Optimal hyperparameter settings for machine learning algorithms used in injection attack detection.

Hyperparameters	XSS	SQL	OSC	LFI	
max_depth	7	7	8	4	
min_samples_split	62	82	62	42	
min_samples_leaf	7	12	7	12	
cross_validation (cv)	100	100	100	100	
n_iter	10	10	10	10	

Automated testing of the final model was performed using Burp Suite to simulate an attack scenario. The test consisted of a total of 36,269 HTTP requests, replicating real-world traffic patterns. Of these, 18,829 were malicious requests designed to exploit vulnerabilities, while 17,440 were legitimate requests. This simulated attack allowed for the evaluation of the WAF’s effectiveness under a realistic workload.

A confusion matrix (Fig. 9) was utilized to evaluate the performance of the WAF in classifying incoming HTTP requests as either legitimate (0) or potentially malicious (1), indicating vulnerabilities such as XSS, SQL injection, LFI, or OSC injection The confusion matrix reveals the distribution of classification results, with 17,297 true positives (correctly identified attacks), 16,533 true negatives (legitimate requests correctly classified), 907 false positives (benign requests misclassified as attacks), and 1,532 false negatives (missed attacks). These results form the basis for the WAF’s evaluation metrics and indicate areas in need of improvement. The high number of false negatives suggests that obfuscated or infrequent attack patterns may not have been adequately represented in the training data, while false positives likely stem from benign but structurally atypical inputs. Such misclassifications point to the limitations of manual feature engineering in capturing the complexity and diversity of modern web traffic. Moreover, the real-time data collection process predominantly targeted specific input fields, which constrained the variability of attack vectors and may have hindered the model’s ability to generalize across broader application contexts. Expanding the dataset to include more diverse input structures and targeting a wider range of application components is expected to improve the model’s robustness, reduce misclassification rates, and enhance performance in dynamic real-world environments.

Figure 9 Confusion matrix of the proposed anomaly-based waf evaluated on simulated data for detecting XSS (A), SQL injection (B), LFI (C), and OS command injection (D) attacks.

Accuracy (0.9327) was determined using Eq. (3), measuring the proportion of correctly classified instances across all requests. Precision (0.9151), calculated by Eq. (4), assessed the reliability of positive predictions by determining the proportion of actual threats among all detected threats. Recall (0.9479), derived from Eq. (5), evaluated the model’s effectiveness in detecting attacks by computing the proportion of correctly identified threats relative to the total number of actual threats. F1-score (0.9313), obtained using Eq. (6), provided a balanced measure by incorporating both precision and recall, mitigating the impact of false positives and false negatives.

Interpreting the confusion matrix is crucial for understanding the WAF’s detection capabilities beyond a singular accuracy metric. The relatively low number of false positives and false negatives indicates a strong ability to minimize misclassification errors, reinforcing the model’s efficacy in identifying malicious activity. However, the presence of false positives may lead to the unnecessary blocking of legitimate requests, while false negatives highlight undetected threats, emphasizing the inherent trade-offs in web security applications.

In many studies on the protection of web applications, the performance of proposed methods has been evaluated primarily in simulation environments, without testing on dynamic web traffic. To ensure the validity of the results, evaluations must also be conducted in live network conditions.

Table 7 compares studies that conducted live testing with those based on simulation environments, along with the performance results of our proposed model under both conditions. Unlike prior work, our model is validated in both simulated and live environments, demonstrating that its detection capability remains reliable under practical, real-world scenarios.

Table 7 Comparison of the proposed waf model with existing studies in terms of detection performance under simulated and real-time conditions.

	References	Best ML algorithm	Dataset	XSS	SQL	OSC	LFI	Anomaly detection accuracy	
	Isiker & Sogukpinar (2021)	SVM	CSIC 2010	+	+	+
(limited)	+
(limited)	XSS, SQL, OSC, LFI and others %99.53	
SIMULATION	Guven et al. (2022)	RF	CICIDS2017	+	+	–	–	XSS, SQL and others %99.94	
	Shaheed & Kurdy (2022)	NB	CSIC 2010, HTTPParams-2015, their data	+	+	+
(limited)	+
(limited)	98.8%
XSS, SQL, OSC, LFI	
	Proposed	DT	CSIC 2010, HTTPParams-2015, our data	99.65%
accuracy	99.57%
accuracy	98.18%
accuracy	93.81%
accuracy	No average accuracy	
	Arjunan (2024)	KNN	NSL-KDD, ISCX-IDS, and
CTU-13	+	+	+
(limited)	+
(limited)	78.1% DOS, DDOS and other	
REAL-TIME	Dawadi, Adhikari & Srivastava (2023)	LSTM	ISCX, CISC 2010 and CICDDoS	+	+	+
(limited)	+
(limited)	97.57% DDOS
89.34% SQL, XSS	
	Proposed (Real-time)	DT	CSIC 2010, HTTPParams-2015, our data	+	+	+	+	93.27%	
Notes.

+ , −: indicate whether the respective study includes the detection of XSS, SQL injection, OSC, and LFI attacks.limited: restricted number of samples.

To identify the optimal ML approach for our WAF, we first assessed the detection performance of various traditional algorithms (KNN, LR, NB, SVM, DT) for the targeted attacks in a simulated environment. The results were comparable to those reported in prior studies, such as Isiker & Sogukpinar (2021), Guven et al. (2022), and Shaheed & Kurdy (2022). After selecting the best-performing model, we applied it in a dynamic test environment, where our results surpassed those of Arjunan (2024) and Dawadi, Adhikari & Srivastava (2023), which focus on real-time web application attack detection. This demonstrates the efficacy of our approach in live conditions.

A critical limitation of many existing studies is their reliance on simulated environments and outdated or narrow datasets, which do not fully reflect the complexity or diversity of modern web attacks. Our study addresses these gaps by combining live network traffic with benchmark datasets, creating a hybrid dataset that is both diverse and up-to-date. Moreover, whereas most prior models focus primarily on XSS and SQL injection, our approach also includes LFI and OSC injection, enhancing its coverage and relevance.

Compared to previous work, the proposed model introduces distinct advantages in terms of dataset realism, broader attack coverage, and real-time evaluation methodology. While many studies use benchmark or hybrid datasets, few evaluate their models under live traffic conditions or validate them in dynamic web contexts. By conducting thorough testing in real network environments and incorporating systematic feature selection and a wide range of attack types, our model demonstrates both practical applicability and novelty in securing modern web applications.

Conclusions

In this study, a ML-based WAF was developed to protect web applications against XSS, SQL injection, LFI, and OSC injection. The model utilizes a DT classifier trained on a hybrid dataset that combines publicly available benchmark datasets (CSIC2010, HTTPParams 2015) with live attack traffic collected via Burp Suite. This dataset improves detection accuracy by addressing the limitations of outdated benchmark-only training.

The experimental findings indicate that the proposed WAF achieves a high level of accuracy in detecting injection attacks, with an F1-score of 93.13% and an accuracy of 93.27% during live evaluation (Q1). These results confirm that traditional ML models, particularly the DT classifier, can effectively identify injection attacks under operational conditions, offering a viable alternative to rule-based WAFs. Moreover, the performance of the proposed model is comparable to, and in some cases exceeds that of recent studies on active web attack detection (e.g., Arjunan, 2024; Dawadi, Adhikari & Srivastava, 2023), further emphasizing its practical applicability (Q3). The WAF operates in a live environment, ensuring continuous security monitoring and adaptive threat detection without relying on manually updated rule-based systems.

The research aimed to evaluate the effectiveness of ML-based WAFs in detecting web-based injection attacks in live environments. The results demonstrate that the DT-based WAF effectively mitigates injection attacks, confirming its suitability for deployment in dynamic traffic scenarios (Q2).

The developed WAF offers several advantages for web application security. By leveraging ML, it can dynamically adapt to evolving attack patterns, reducing dependence on static rule-based security measures. The real-time nature of the system allows for immediate detection and mitigation of threats, making it a viable solution for deployment in dynamic web environments. Furthermore, integrating this WAF into DevSecOps pipelines supports a shift-left security approach, enabling early detection and reducing the need for post-deployment security patches.

Despite its effectiveness, the proposed WAF has certain limitations. The model’s performance depends on the quality and diversity of its training data. While the dataset improves detection accuracy, incorporating additional real-world attack data-especially as new threats emerge-could further enhance robustness. Additionally, like most ML-based security models, the DT classifier may be vulnerable to adversarial attacks, where manipulated inputs evade detection. Future research should explore ensemble learning or deep learning approaches to improve resilience against such attacks. Another limitation is the focus on a specific subset of injection attacks. Expanding coverage to include other OWASP Top 10 vulnerabilities would increase the WAF’s applicability. Lastly, testing the model in diverse real-world deployment scenarios, including different network conditions and web environments, will provide deeper insights into its scalability and practical effectiveness. By addressing these challenges, future iterations of this WAF can further strengthen web application security and contribute to the ongoing development of ML-driven cybersecurity solutions.

Additional Information and Declarations

Competing Interests

Author Contributions

Data Availability

The authors declare there are no competing interests.

Muhammed Ersin Durmuşkaya conceived and designed the experiments, performed the experiments, analyzed the data, performed the computation work, prepared figures and/or tables, authored or reviewed drafts of the article, and approved the final draft.

Selim Bayraklı conceived and designed the experiments, authored or reviewed drafts of the article, and approved the final draft.

The following information was supplied regarding data availability:

Data is available at figshare:

DURMUŞKAYA, Muhammed Ersin; DURMUŞKAYA, MUHAMMED ERSİN; BAYRAKLI, Selim (2024). ML Based WAF-mitmproxy: csv and log files. figshare. Dataset. https://doi.org/10.6084/m9.figshare.26583526.v1

The CSIC 2010 dataset available at GitHub: https://github.com/aref2008/waf/blob/master/CSIC.csv

The HTTPParams 2015 dataset available at GitHub: https://github.com/Morzeux/HttpParamsDataset

Codes are available at GitHub and Zenodo:

https://github.com/ersindurmuskaya/ml-based-waf-mitmproxy-tool/tree/main/waf/3_webserver_log_files

ersindurmuskaya. (2025). ersindurmuskaya/ml-based-waf-mitmproxy-tool: Web Application Firewall Based on Machine Learning Models (v1.0.0). Zenodo. https://doi.org/10.5281/zenodo.15574488.

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
