# Peer review of "Web application firewall based on machine learning models"

_PeerJ Computer Science, doi:10.7717/peerj-cs.2975_

## Round 0.1 · original submission · Major Revisions

Please address all the comments of the reviewers in a revision

Reviewer 1 ·

Basic reporting

1. Language and Clarity:
- The manuscript is generally written in clear and professional English. However, there are occasional grammatical and typographical errors (e.g., "enjection" in the keywords). I suggest a thorough proofreading to eliminate such errors and ensure consistency in technical terms.
- The text includes some unnecessarily complex sentences, which could be simplified for better readability. For instance, sentences in the abstract and introduction should be revised to avoid redundancy while maintaining precision.

2. Introduction and Background:
- Generally speaking, the authors have outlined the motivation for this study. However, they miss some important and related works to the studied topic, i.e., machine learning empowered Web application firewalls (WAF) and software engineering. More specifically, to the best of the reviewer's knowledge, https://dl.acm.org/doi/10.1145/3319619.3322026 is the first work in the literature that defines the test case generation problem for WAF as a search problem and used semantic learning to navigate the effective search. Later, https://ieeexplore.ieee.org/abstract/document/10372386 extends the previous work to a multi-task scenario beyond WAF. In terms of security and robustness of software engineering problems using genetic algorithms and machine learning, the authors should consider giving credit to https://dl.acm.org/doi/10.1145/3660808, https://dl.acm.org/doi/10.1145/3520304.3528981, https://link.springer.com/chapter/10.1007/978-3-031-14714-2_24, and https://arxiv.org/abs/2403.13588.

3. Structure and Format:
- The manuscript generally follows the standard structure of academic articles. Sections such as the abstract, introduction, methodology, results, and conclusions are well-defined. However, some subsections (e.g., "Data Set") could be merged into the methodology for better cohesion.
- Figures are relevant and well-labeled, but their resolution appears inconsistent in some cases. For instance, the ROC curve in Figure 7 could be clearer to improve interpretability.

4. Self-Contained and Unit of Publication:
- The submission is largely self-contained and represents a significant unit of publication. However, the conclusions section could elaborate further on practical implications and limitations to ensure the study stands alone as a comprehensive resource.

5. Formal Results and Definitions:
- Key terms and methodologies are well-defined. The mathematical formulas provided (e.g., for precision, recall, and F1-score) are clear and necessary for replicating the results.
- The manuscript would benefit from a table summarizing the comparative performance of the proposed WAF model against other state-of-the-art approaches.

Experimental design

See "Basic reporting".

Validity of the findings

See "Basic reporting".

Additional comments

N/A

Cite this review as

Reviewer 2 ·

Basic reporting

- Language and Clarity: The article generally uses clear and professional English; however, there are a few instances where phrasing could be simplified for better clarity. For example, consider revising overly complex sentences to enhance readability.
- Literature References: While the literature is adequately referenced, some key studies in the field of machine learning and web application firewalls could be included to provide a more comprehensive background. Including recent advances or contrasting approaches would strengthen the context.
- Article Structure: The structure is mostly professional; however, it would benefit from a clearer separation of sections. For instance, subheadings within the methodology could provide better navigation and understanding of the experimental setup.
- Figures and Tables: The figures and tables are well-presented but some lack detailed captions that would help the reader understand the data being represented without needing to refer back to the text. Consider adding more descriptive captions.
- Self-Containment: The results presented are relevant to the hypotheses, but further elaboration on the implications of the findings would enhance the discussion. Ensure that all technical terms are defined clearly for accessibility to readers from different backgrounds.
- Formal Results: The definitions of key terms are sometimes vague. It is important to provide clear definitions for all terms used and ensure that proofs or justifications for the results are detailed enough for reproducibility.

Suggested Improvements:

- Simplify complex sentences for clarity.
- Include more recent literature to enhance background context.
- Improve section headings and captions for better navigation and understanding.
- Ensure all technical terms are well-defined and any claims are supported by detailed explanations.

Experimental design

- Research Relevance and Knowledge Gap: The research question is well-defined and addresses a significant knowledge gap in the field of web application security. However, a more explicit statement regarding how this research specifically fills the identified gap would enhance clarity. Consider adding a brief discussion on the uniqueness of your approach compared to existing studies.
- Technical and Ethical Standards: The investigation appears to adhere to high technical standards, but additional information about ethical considerations, especially regarding data collection and usage, would strengthen the paper. Explicitly mentioning any ethical approvals or guidelines followed would be beneficial.
- Methodological Detail: While the methods are generally described, there are areas where more detail is needed. Specifically, the parameters used in the machine learning algorithms should be outlined more comprehensively. This includes any preprocessing steps, hyperparameter settings, and the rationale behind the choice of algorithms.
- Replication: To ensure the study is replicable, it would be helpful to include more specifics about the dataset, such as how it was constructed and any challenges encountered during data collection. Additionally, providing code snippets or links to the implementation would support reproducibility.

Suggested Improvements:

- Clarify how the research fills the identified knowledge gap in more detail.
- Include information about ethical considerations related to data usage.
- Provide more comprehensive details on methodology, including algorithm parameters and preprocessing steps.
- Enhance replication potential by detailing dataset construction and providing access to implementation resources.

Validity of the findings

- Assessment of Impact and Novelty: The article does not sufficiently assess the impact and novelty of the findings. A clearer discussion on how the results contribute to existing literature and their potential implications for practice would enhance the paper. Consider explicitly stating the novel aspects of your research compared to previous studies.
- Replication of Findings: While the findings are promising, the article would benefit from a discussion on the potential for replication. Highlighting the rationale for why replication studies are important in this context and suggesting how future research could build upon your work would strengthen the argument for the validity of your findings.
- Robustness of Data: The underlying data appear to be robust; however, a more detailed analysis of the statistical methods used to validate the findings would enhance confidence in the results. Including confidence intervals, p-values, or other statistical metrics would provide a clearer picture of the data's reliability.
- Conclusions: The conclusions are generally well stated and linked to the original research question. However, they could be further refined by explicitly summarizing how each key finding supports the conclusions drawn. Avoid introducing new information in the conclusion; instead, ensure it strictly reflects the results presented.

Suggested Improvements:

- Add a discussion on the impact and novelty of the findings, emphasizing their contribution to the field.
- Encourage consideration of replication studies and outline the rationale for their importance.
- Include more detailed statistical analysis to reinforce the robustness of the data.
- Refine the conclusion to ensure it strictly summarizes findings without introducing new concepts.

Cite this review as

·

Basic reporting

Enhance Clarity and Readability: The manuscript should be revised for clarity and readability. Some sections contain complex sentences that may confuse readers. Simplifying the language and breaking down long sentences can help convey the message more effectively.

Expand the Introduction: The introduction should provide a more detailed background on the topic, including a clearer explanation of the significance of the research. Adding context about the current state of web application firewalls and their challenges will help readers understand the importance of the study.

Strengthen Literature Review: The literature review could benefit from a more comprehensive analysis of recent studies related to web application firewalls and machine learning. Including a broader range of references will demonstrate a thorough understanding of the field and highlight how this research contributes to existing knowledge.

Improve Figure and Table Descriptions: Ensure that all figures and tables are accompanied by detailed captions that explain their relevance to the study. This will help readers interpret the data more easily and understand how it supports the research findings. Additionally, check that all visual elements are of high quality and appropriately formatted.

Experimental design

Detailed Methodology Description: The methodology section should provide a more comprehensive description of the experimental setup, including specific details about the datasets used, the selection criteria for these datasets, and the preprocessing steps taken. This will enhance the reproducibility of the study.

Clear Definition of Research Questions: The research questions should be explicitly stated and clearly defined. This will help readers understand the objectives of the study and how the experimental design addresses these questions. Consider framing the research questions in a way that highlights their relevance to the field.

Justification of Chosen Methods: The article should include a justification for the choice of machine learning algorithms and techniques used in the study. Discussing why specific methods were selected over others will provide insight into the decision-making process and strengthen the validity of the findings.

Evaluation Metrics: The experimental design should specify the evaluation metrics used to assess the performance of the machine learning models. Clearly defining metrics such as accuracy, precision, recall, and F1-score will allow for a better understanding of how the results are measured and compared to other studies in the field.

Validity of the findings

tatistical Analysis and Robustness: The manuscript should include a more detailed statistical analysis to support the validity of the findings. It is important to provide information on the statistical tests used, the significance levels, and any confidence intervals for the results. This will help establish the robustness of the conclusions drawn from the data.

Cross-Validation and Generalizability: The study should discuss the use of cross-validation techniques to assess the generalizability of the machine learning models. Providing details on how the models were validated (e.g., k-fold cross-validation) and the results of these validations will enhance the credibility of the findings and demonstrate that the models are not overfitting to the training data.

Additional comments

The manuscript should include a comparison of the proposed machine learning models with baseline models or existing methods in the literature. This comparison will help to contextualize the performance of the new models and demonstrate their effectiveness relative to established approaches, thereby strengthening the validity of the findings.
The study should acknowledge and address any potential biases in the dataset used for training and testing the models. Discussing how the dataset was curated, including any limitations or imbalances, will provide transparency and help readers assess the potential impact of these biases on the validity of the findings.
I recommend that the authors provide a more comprehensive discussion of their findings in relation to existing literature to clarify their contributions to the field.

Cite this review as

---

## Round 0.2 · Minor Revisions

One of the reviewers raised a number of issues that need to be addressed.

Reviewer 1 ·

Basic reporting

The authors have successfully addressed my concerns. Now it is acceptable.

Experimental design

The experimental design has been improved.

Validity of the findings

The findings are updated.

Cite this review as

Reviewer 4 ·

Basic reporting

Language and Clarity: The article is well structured and written in clear English, however, some repetitive phrases need to be revised and could be simplified to make the article easier to read.

Literature references, sufficient field background/context provided: From my point of view, the study of related works and the bibliography are correct.

Self-contained with relevant results to hypotheses: The results obtained are adequately related to the hypotheses raised, and to the conclusions shown.

Formal results: In the latest version of the paper the definitions are expanded, however, although they are well known, perhaps a short introduction to the algorithms used would be relevant, so that any reader can have a better understanding of the paper.

Experimental design

Original primary research within Aims and Scope of the journal: The research questions are well defined and address a topic of great relevance to WAF security.

Rigorous investigation performed to a high technical & ethical standard: The latest version of the article adds a paragraph on ethical compliance, which is important considering the subject matter, knowing that no unauthorized access or intrusions were performed on real systems.

Methods described with sufficient detail & information to replicate: The methods are described in a general way and in my opinion there are aspects such as the ML methods themselves and their parameters that need a greater degree of explanation. The dataset and the code used are accessible, which allows replication of the experimentation.

Evaluation metrics: The latest version of the article defines the purpose of the metrics used, however further justification of their choice based on the algorithms used would be advisable.

Validity of the findings

Impact and novelty not assessed: It would be necessary to expand on the novelty of this work with respect to those proposed in the literature review, as well as to highlight the impact of this work.

Conclusions: In the latest version of the article, the conclusions are well structured, formulated and linked to the original research question.

Additional comments

The introduction section clearly and sequentially presents the fundamental concepts necessary to understand the work, including IDS, IPS, WAFs, common vulnerabilities, and the limitations of traditional solutions. The writing is accurate and coherent, employing appropriate technical language. This section successfully fulfills its purpose by delineating the problem, justifying the need for new solutions, and presenting the research questions guiding the study. The broader coverage of attack types compared to traditional studies highlights the depth of the work. However, it would be beneficial to provide a more precise explanation of the differences between traditional machine learning techniques and deep learning approaches, and their respective suitability for this study. The contributions are clearly enumerated. Finally, the article organization section contains a minor typographical error ("follow" should be "follows").

The Related Work section presents an extensive and updated review of relevant research on WAFs, their evolution towards machine learning methods, and attack evaluation techniques. The writing is clear, technical, and appropriate for the scientific context, although there are some issues regarding organization and style. The section presents a collection of studies; however, it would be advisable to incorporate a comparative analysis between them rather than adopting a purely descriptive approach. In terms of technical depth, the section successfully offers a broad overview of recent and relevant studies, covering not only attack detection but also advancements in automated testing techniques and evolutionary strategies. Nonetheless, a brief critical analysis of the limitations of the reviewed works would strengthen the justification for the proposed model.

The Materials and Methods section provides a detailed description of the implementation, training, and evaluation process of the proposed model, demonstrating methodological rigor in the construction of the hybrid dataset, the feature extraction process, and the application of statistical techniques for feature selection. The combined use of Chi-Square, PMI, and Mutual Information reinforces the validity of the selected features for the machine learning models. It is also positive that standard practices such as data normalization, standardization, and 10-fold cross-validation are applied; however, a more thorough justification of the evaluation metrics used is necessary.

The Results section presents in detail the experimental setup, model hyperparameters, and evaluation metrics used. It demonstrates a solid approach to model validation, employing robust practices such as 10-fold cross-validation and hyperparameter optimization through RandomizedSearchCV. The results analysis is comprehensive, including both classification metrics and practical evaluations through real traffic simulations using Burp Suite. Additionally, an explicit comparison between performance in simulated environments and real-time traffic is conducted, adding a practical dimension that many studies omit. However, a deeper analysis of the model's errors and a more critical comparison with other works discussed in the Related Work section would be desirable.

The Conclusions section clearly summarizes the main contributions of the study. The writing is generally fluent and well-structured around the research questions (Q1, Q2, Q3), thereby reinforcing coherence with the initial objectives. It is appropriate to acknowledge that the model’s performance depends on the quality of the dataset and that there remains a risk from adversarial attacks, which reflects a mature analysis. Furthermore, it is positive that clear directions for future research are proposed, such as integrating ensemble learning techniques and expanding coverage to additional OWASP Top 10 vulnerabilities.

Cite this review as

---

## Round 0.3 · accepted · Accept

The authors smoothly addressed the points raised by the reviewers and therefore I can recommend this manuscript for acceptance and publication.